# Discrete Element Simulation of Interaction between Hydraulic Fracturing and a Single Natural Fracture

**Rouhollah Basirat, Kamran Goshtasbi *** and **Morteza Ahmadi**

Rock Mechanics Engineering Division, Faculty of Engineering, Tarbiat Modares University, Tehran 14115-111, Iran; r.basirat@modares.ac.ir (R.B.); moahmadi@modares.ac.ir (M.A.)

* Correspondence: goshtasb@modares.ac.ir

**Abstract:** Hydraulic fracturing (HF) treatment is performed to enhance the productivity in the fractured reservoirs. During this process, the interaction between HF and natural fracture (NF) plays a critical role by making it possible to predict fracture geometry and reservoir production. In this paper, interaction modes between HF and NF are simulated using the discrete element method (DEM) and effective parameters on the interaction mechanisms are investigated. The numerical results also are compared with different analytical methods and experimental results. The results showed that HF generally tends to cross the NF at an angle of more than 45° and a moderate differential stress (greater than 5 MPa), and the opening mode is dominated at an angle of fewer than 45°. Two effects of changing in the interaction mode and NF opening were also found by changing the strength parameters of NF. Interaction mode was changed by increasing the friction coefficient, while by increasing the cohesion of NF it was less opened under a constant injection pressure.

**Keywords:** hydraulic fracturing; natural fracture; fracture network; interaction modes

## 1. Introduction

Hydraulic fracturing (HF) is a widely used well-stimulation technique for enhancing the productivity in the petroleum industry. This technique is applied in low-permeability reservoirs even in the fractured reservoirs. Extensive field observations showed that many reservoirs contain preexisting natural fractures (NF). In addition, the characterization of the fracture network for evaluating fracturing operations and for predicting reservoir performance is a common challenge for petroleum-fractured reservoirs. Characterizing stimulated fractured reservoirs is challenging because of complexities such as dual-porosity effects, multiphase flow, complex flow regimes, complex fracture geometry, the role of natural fractures, liquid loading, and operational complexities [1,2]. There are some methods for characterizing the fracture network. The most common methods for fracture characterization are pressure-transient analysis, rate-transient analysis, microseismic analysis, and tracer test. These methods are widely used by industries, and each has its own pros and cons which is discussed by Zolfaghari et al. [3].

The NF may result in a substantial difference in the HF propagation geometry. The interaction between HF and NF is a complex process. When HF intersects a NF, it can cross or be arrested by the NF; and the HF can subsequently open (dilate) the NF (Figure 1). First, the fracture tip reaches the interface, but the fluid front remains further back due to the fluid lag. At this moment, the net fluid pressure (the difference between the fracturing fluid pressure and the minimum in situ stress) at the intersection point can be considered zero, but the natural fracture is already under the influence of the stress field generated by the hydraulic fracture. This step can be analyzed by the mechanical interaction between the hydraulic fracture and the natural fracture without considering fluid flow. There are two possible outcomes from this interaction: one is slippage or arrest (Figure 1b), and the other is crossing

(Figure 1c). Shortly thereafter, the second step in the process occurs when the fluid front reaches the natural fracture, and fluid pressure at the intersection point rises. In the case of slippage, the fluid may flow into and dilate the natural fracture if the fluid pressure is larger than the normal compressive stress on the natural fracture. If the flow continues, the dilated natural fracture becomes part of a hydraulic fracture network (Figure 1d); i.e., the hydraulic fracture turns and propagates along the natural fracture [4].

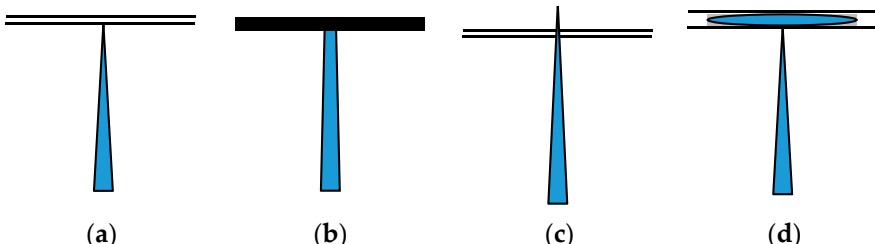

**Figure 1.** A schematic of different interaction modes between hydraulic fracturing (HF) and natural fracture (NF). (**a**) Approaching HF to NF. (**b**) Arresting. (**c**) Crossing. (**d**) Opening.

These interaction modes depend on the in situ stresses, the orientation of induced fracture with respect to the NF, mechanical properties of the rock, properties of NF, and the HF treatment parameters, including fracturing fluid properties and injection rate. In this regard, several works have been published in three categories of experimental [5–12], analytical [13–16], and numerical methods [17–23].

Warpinski et al. [5,6] found that the growth and geometry of hydraulic fractures were affected by the stress, joints, faults, interfaces and layer material property. Much research has studied factors affecting HF with the development of laboratory equipment and construction of True-Triaxial test devices [7–12]. Zhou et al. [7] also studied the interaction between HF and NF through a series of experiments and presented the crossed boundary based on different stress regimes. Sarmadivaleh et al. [8] investigated the influences of interface cohesion and approach angle on the interaction between NF and HF. Dehghan et al. [9,10] studied the effect of dip direction of NF on HF propagation as well as stress state and approach angle. Guo et al. [11] studied several factors including the type of injecting fracturing fluids, pump flow rate, fracturing pressure curve characteristics, and fracture morphology. Chen et al. [12] focused on the effects of the notch angle, notch length, and injection rate on hydraulic fracturing.

Many studies have been conducted on the effect NF on HF using different numerical methods.

Dahi-Taleghani and Olson [17] presented a numerical model based on the extended finite-element method (XFEM) as a design tool that can be used to optimize treatment parameters under complex propagation conditions. Sesetty and Ghassemi [18] presented a boundary element-based method for modeling the interaction modes. Their results showed the complexity of the propagation process and its impact on stimulation design and proppant placement. Behina et al. [19,20] studied the mechanical activation of a NF because of the propagation of the HF using displacement discontinuous method (DDM). Their results displayed before the HF reaches a NF, and the fracture re-initiation across the NF and with an offset is probable. Liu et al. [21] presented a fully coupled finite-element method (FEM)-based hydraulic-geomechanical fracture model accommodating gas sorption and damage to better understand the interaction between hydraulic fracture and oriented perforation. Liu et al. [22] developed a hybrid finite volume and XFEM for simulating hydro-fracture propagation in quasi-brittle materials. Han et al. [23] adopted a modified fluid-mechanically coupled algorithm in the Particle Flow Code (PFC$^{2D}$) to study the influence of grain size heterogeneity and in situ stress on hydraulic fracturing behavior. They found that the initiation and breakdown pressure are gradually reduced with the increase of the grain size heterogeneity.

The above brief introduction indicates the complexity of the interaction between HF and NF. In order to understand the interaction mechanism and the importance of each parameter, it is necessary to conduct a fundamental study on simple cases first. Hence, in this paper, a single NF with different

orientations is considered for investigating the interaction mechanism by changing the state of in situ stresses, approach angle, and NF strength properties (which is the weakness of previous research). To achieve this aim, more than 60 numerical models were simulated using the discrete element method (DEM). Finally, the numerical results were compared and discussed with analytical methods.

## 2. Analytical Methods

Various authors have provided analytical equations and numerical solutions for predicting the interaction between a natural fracture and hydraulic fracture. Blanton [13] derived fracture interaction criteria relating differential stress and angle of approach. Renshaw and Pollard [14] provided a criterion for crossing unbonded interfaces. Sarmadivaleh and Rasouli [16] developed a criterion for non-orthogonal cohesive fracture. In this section, these analytical criteria are presented.

Blanton [13] presented a simple analytical fracture interaction criterion on differential stress and angle of interaction to extrapolate the laboratory results to field simulations (Figure 2).

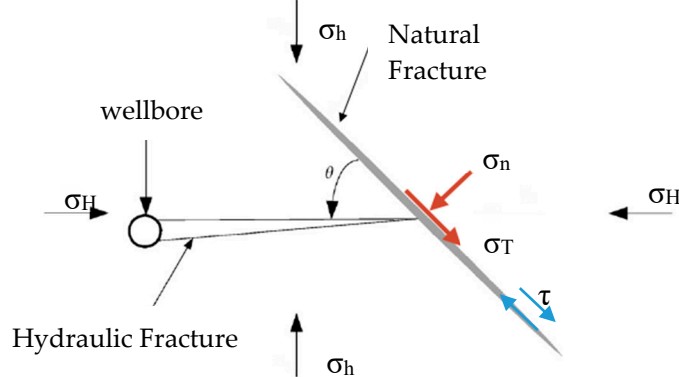

**Figure 2.** A schematic of a HF approaching a NF.

where $\sigma_H$ and $\sigma_h$ are the maximum and minimum principal stress acting on the plane of the NF as shown in Figure 2. $\theta$ is the angle of interaction, and $T_0$ is tensile strength of NF. The criterion is based on the elastic solution of stresses in the interaction zone. The final equation for crossing is given by Blanton as [13]:

$$\frac{(\sigma_H - \sigma_h)}{T_0} < \frac{-1}{\cos 2\theta - b \sin 2\theta} \tag{1}$$

When no slippage zone or no-slip occurs, $b$ approaches infinity which term $b$ tends to:

$$b_\infty = \frac{1}{2\pi} \ln \left[ \frac{1 + \left(1 + e^{\frac{\pi}{2\mu_f}}\right)^{0.5}}{1 - \left(1 + e^{\frac{\pi}{2\mu_f}}\right)^{0.5}} \right]^2 \tag{2}$$

and $b_\infty$ becomes a function of the NF friction coefficient ($\mu_f$). Blanton used this value for comparison of his analytical and experimental results.
and $b_\infty$ becomes a function of the natural fracture friction coefficient. Blanton used this value to compare his analytical and experimental results.

Renshaw and Pollard [13] developed a simple criterion for predicting if a fracture will propagate across a frictional interface orthogonal to the fracture based on the linear elasticity fracture mechanics solution for the stresses near the fracture tip. It determines the stresses required to prevent slip along the interface at the moment when the stress on the opposite side of the interface is sufficient to reinitiate a fracture. Because the approach angle has a significant effect on the crossing, the Renshaw and Pollard criterion were extended to fracture intersection at non-orthogonal angles by Gu and Weng [15]. Later,

Sarmadivaleh and Rasouli [16] developed Renshaw and Pollard criterion in the form a single analytical formula for a non-orthogonal cohesive fracture as the following formula:

$$\frac{-\sigma_n}{(T_0 - \sigma_T)} < \frac{\left(1 - \sin\frac{\theta}{2}\sin\frac{3\theta}{2}\right) + \frac{1}{\mu_f'' \cos\frac{\theta}{2}}\left(\left|\sin\frac{\theta}{2}\cos\frac{\theta}{2}\cos\frac{3\theta}{2} + \alpha\right|\right)}{\left(1 + \sin\frac{\theta}{2}\sin\frac{3\theta}{2}\right)} \tag{3}$$

where $\sigma_n$ and $\sigma_T$ are the normal and shear stress acting on the NF as shown in Figure 2, respectively. In the Equation (3), $\mu_f'' = \mu_f' + \mu_f$, and $\mu_f'$ and $\alpha$ can be calculated using:

$$\mu_f' = \frac{\frac{\tau_0}{\sigma_n}}{\left(\frac{\left(1 - \sin\frac{\theta}{2}\sin\frac{3\theta}{2}\right)}{\left(1 - \sin\frac{\theta}{2}\sin\frac{3\theta}{2}\right) + \frac{1}{\mu_f \cos\frac{\theta}{2}}\left|\left(\sin\frac{\theta}{2}\cos\frac{\theta}{2}\cos\frac{3\theta}{2} + \alpha\right)\right|} - 1\right)} \tag{4}$$

$$\alpha = \frac{\tau}{\frac{T_0 - \sigma_T}{\cos\frac{\theta}{2}\left(1 + \sin\frac{\theta}{2}\sin\frac{3\theta}{2}\right)}} \tag{5}$$

where $\tau_0$ is the shear strength of NF. Transformation of in situ stresses on the plane of NF is illustrated in Figure 2. Shear ($\tau$), tangential ($\sigma_T$), and normal stresses ($\sigma_n$) applied along the interface can be calculated from [24]:

$$\tau = -\frac{\sigma_H - \sigma_h}{2}\sin(\pi - 2\theta) \tag{6}$$

$$\sigma_n = \frac{\sigma_H + \sigma_h}{2} + \frac{\sigma_H - \sigma_h}{2}\cos(\pi - 2\theta) \tag{7}$$

$$\sigma_T = \frac{\sigma_H + \sigma_h}{2} - \frac{\sigma_H - \sigma_h}{2}\sin(\pi - 2\theta) \tag{8}$$

## 3. Numerical Modeling

### 3.1. Discrete Element Method (DEM) for Simulating Hydraulic Fracturing (HF)

In this paper, DEM is used for investigating the effect of a single fracture on HF propagation. The DEM method was originated by Cundall [25] and gradually evolved to the Universal Distinct Element Code (UDEC) and 3 Dimensional Distinct Element Code (3DEC) for solving 2D and 3D problems, respectively. The mechanical interaction between blocks is captured by a compliant contact model that accommodates virtual "interpenetrations" governed by assumed finite stiffnesses to derive normal and tangential contact forces. In this model, the increments of normal stress $\Delta\sigma_n$ and the tangential stress $\Delta\tau$ can be calculated from the relative motion of bonded blocks, and are given by [26]:

$$\Delta\sigma_n = -K_n \Delta u_n \tag{9}$$

$$\Delta\tau = -K_s \Delta u_s \tag{10}$$

where $K_n$ and $K_s$ are the normal and shear stiffness, respectively; $\Delta u_n$ and $\Delta u_s$ are the increments of normal displacement and shear displacement in a time step, respectively.

The DEM approach is able to capture the stress-strain characteristics of intact rocks, the opening/shearing of pre-existing fractures and the interaction between multiple blocks and fractures. Combined with discrete fracture network (DFN) models, it has been widely applied to study the mechanical behavior of fractured rocks. This is a rule-based approach which bypasses many of the complex numerical, geometrical and practical challenges of numerical methods [27].

Since *UDEC* can model fracture propagation along the predefined planes only, all potential fracture pathways must be pre-defined. To incorporate this limitation and provide added degrees of freedom for fracture propagation, a Voronoi tessellation scheme can be used to generate randomly

sized polygonal blocks [28]. Therefore, the Voronoi elements with equivalent parameters of intact rock "i.e., (cohesion, φ, tension)$_{DFN}$ = (cohesion, φ, tension)$_{intact\ rock}$" as well as appropriate stiffness ($K_n$ and $K_s$) were used in this study.

Mechanical deformation of joint apertures changes conductivity, and the blocks in this assemblage are treated as being impermeable. The cubic law for flow within a planar fracture is used, where the flow rate (*q*) is given by [26]:

$$q = ka^3\frac{\Delta P}{l}; k = \frac{1}{12\mu} \tag{11}$$

where, *k* is a joint conductivity factor, *a* is the contact hydraulic aperture, $\Delta P$ is the pressure difference between the two adjacent domains, *l* is the length assigned to the contact between the domains, and *μ* is fluid dynamic viscosity. Furthermore, the fracture flow is idealized by means of a parallel plate model and cubic law, which disregards tortuosity.

The hydro-mechanical coupling is considered by associating the hydraulic aperture to the mechanical joint stiffness and acting effective normal stresses. During the fluid flow calculation, the increment of fluid pressure ($\Delta P$) in a reservoir is computed from the bulk modulus of fluid ($K_f$), the volume of the domain ($V_d$), the sum of the flow volume for one time step ($\Delta t$), and the volume change of the domain due to mechanical loading according to Equation (12) [26]:

$$\Delta P = \frac{K_f}{V_d}\left(\sum q\Delta t - \Delta V_d\right) \tag{12}$$

Reservoir deformations are caused by the fluid pressure exerted on the surrounding media. This force (*f*) is a product of fluid pressure ($P_f$), the length (*l*) exposed to fluid in a domain, and unit thickness (1 m) in an out-of-plane direction.

Figure 3a–c show the boundary conditions, DFN model (Voronoi Elements), and contact elements related to discrete blocks, respectively. 3871 contacts elements are used in the numerical modeling. The Mohr–Coulomb and Coulomb criteria are considered for behavior model of rock mass and joints, respectively. The fluid viscosity is $1 \times 10^{-3}$ Pa.s. Table 1 illustrates the rock mechanical parameters of fractures network and NF.

**Table 1.** Rock mechanical properties of rock mass, discrete fracture network (DFN), and natural fracture.

| Rock Mass Parameter | Unit | Value |
|---|---|---|
| Elastic Modulus | GPa | 24 |
| Poisson's ratio | - | 0.25 |
| Cohesion | MPa | 2 |
| Friction Angle | Degree | 35 |
| Tension Strength | MPa | 1 |
| **Fracture Parameters** | | **DFN** | **Natural Fracture** |
| Normal Stiffness | GPa/m | 350 | 20 |
| Shear Stiffness | GPa/m | 140 | 10 |
| Cohesion | MPa | 2 | 0 |
| Friction Angle | Degree | 35 | 20 |
| Tension Strength | MPa | 1 | 0.01 |

*3.2. Interaction Modes in DEM Simulation*

3.2.1. Opening Mode

The first interaction scenario usually happens when the approach angle is low. In this case, HF is propagated along the NF, and then it is opened. For instance, the opening mode is shown in Figure 4 with an incidence angle of 45° under low differential stresses (σ1 − σ3 = 1 MPa). The displacement vectors clearly approve the opening mode in this condition.

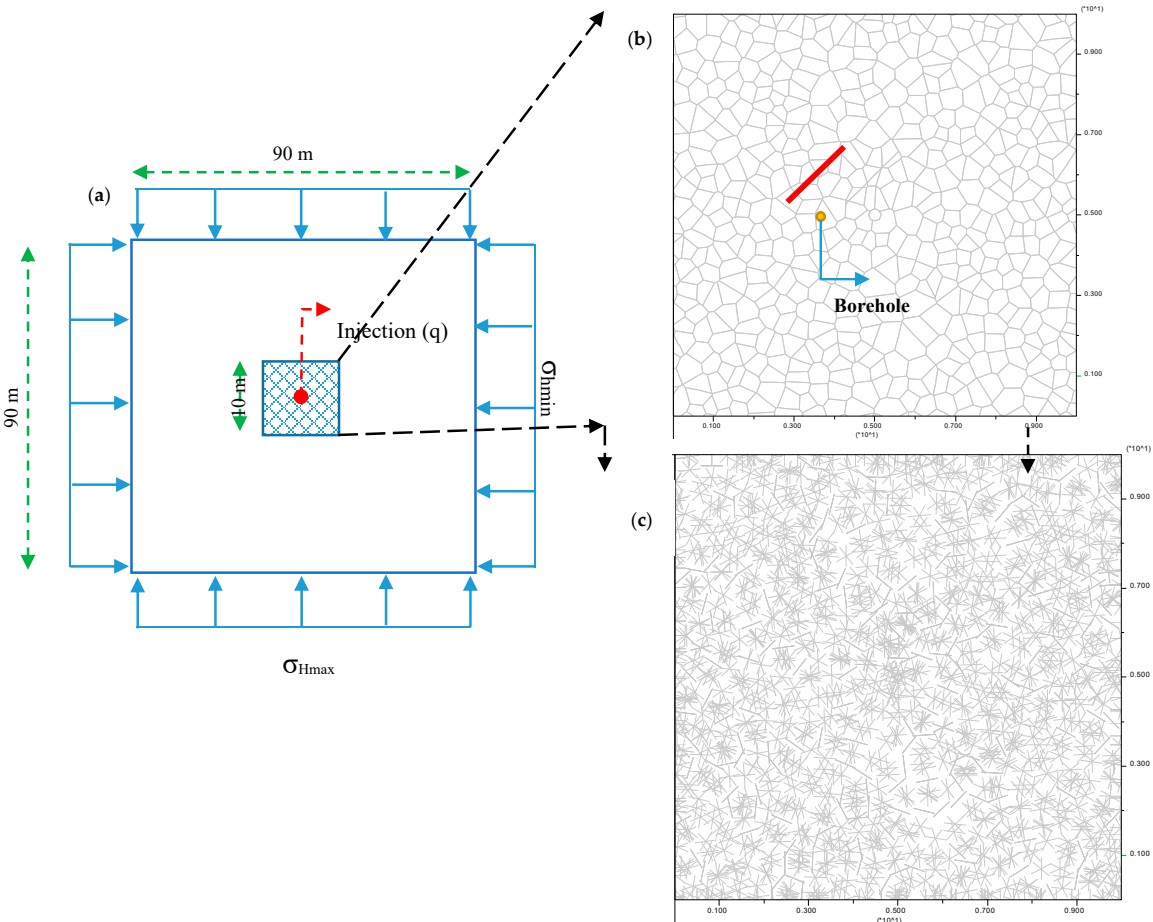

**Figure 3.** (**a**) Dimensions in general numerical mode, (**b**) DFN and NF, (**c**) contact elements location.

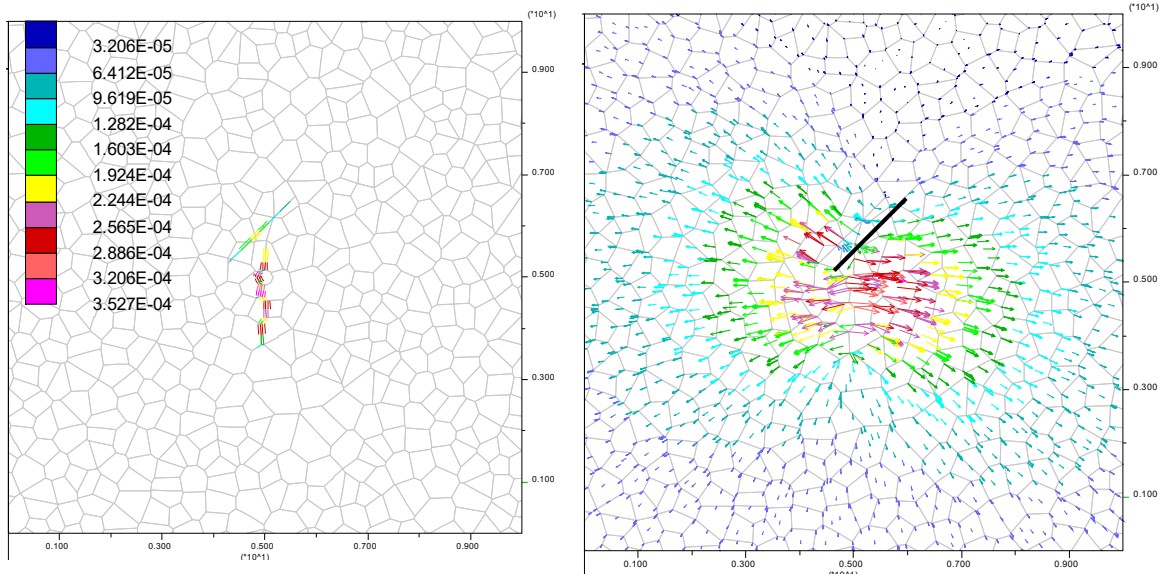

**Figure 4.** Opening mode in numerical modeling with an incidence angle of 45° under low differential stresses of 1 MPa.

### 3.2.2. Arresting Mode

The second interaction scenario indicating an arresting response was observed when the HF approached the NF with an angle of 45° under intermediate differential stress conditions (Δσ = 10 MPa).

Figure 5 shows joint opening and displacement vectors in this condition. In the arresting mode, HF is stopped in the intersection with NF, and then it propagates from the other side of a wellbore with continuing injection during the time (according to Figure 5 and displacement vectors).

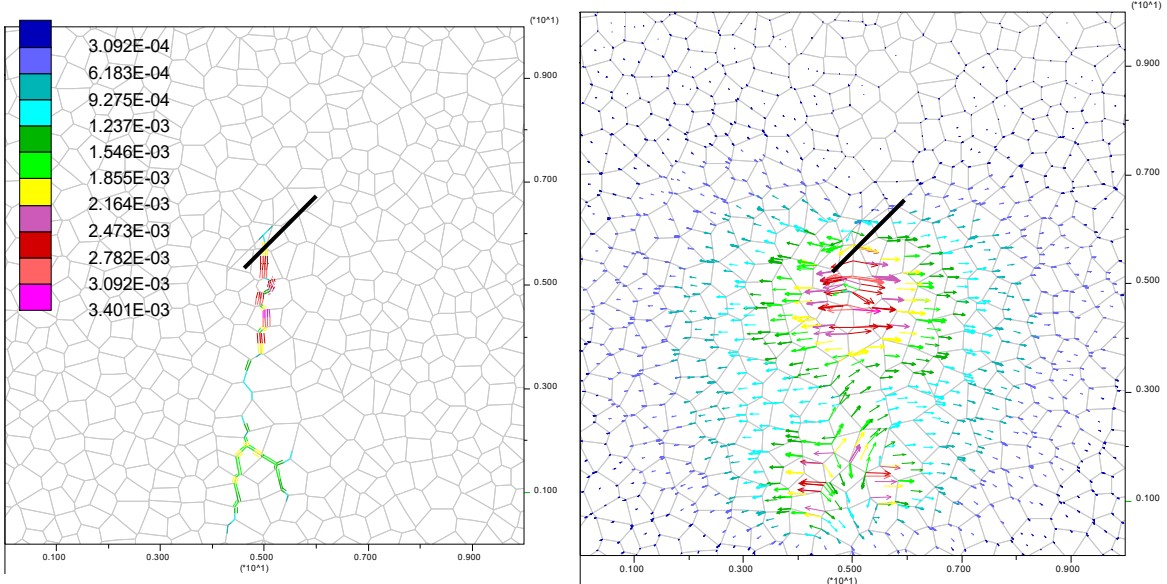

**Figure 5.** Arresting mode in numerical modeling with an incidence angle of 45° under differential stresses of 10 MPa.

### 3.2.3. Crossing Mode

If the HF crosses the NF, it remains planar with the possibility to open the intersected NF if the fluid pressure at the intersection exceeds the normal stress acting on the NF. An example where the numerical modeling produced a hydraulic fracture that crossed a preexisting NF is presented in Figure 6. This figure shows the joint opening and displacement vectors in this condition. In this case, the HF approaches the natural fracture with an incidence angle of 90° under high differential stresses ($\sigma 1 - \sigma 3 = 15$ MPa) and finally crosses the NF along the direction of maximum principal stress.

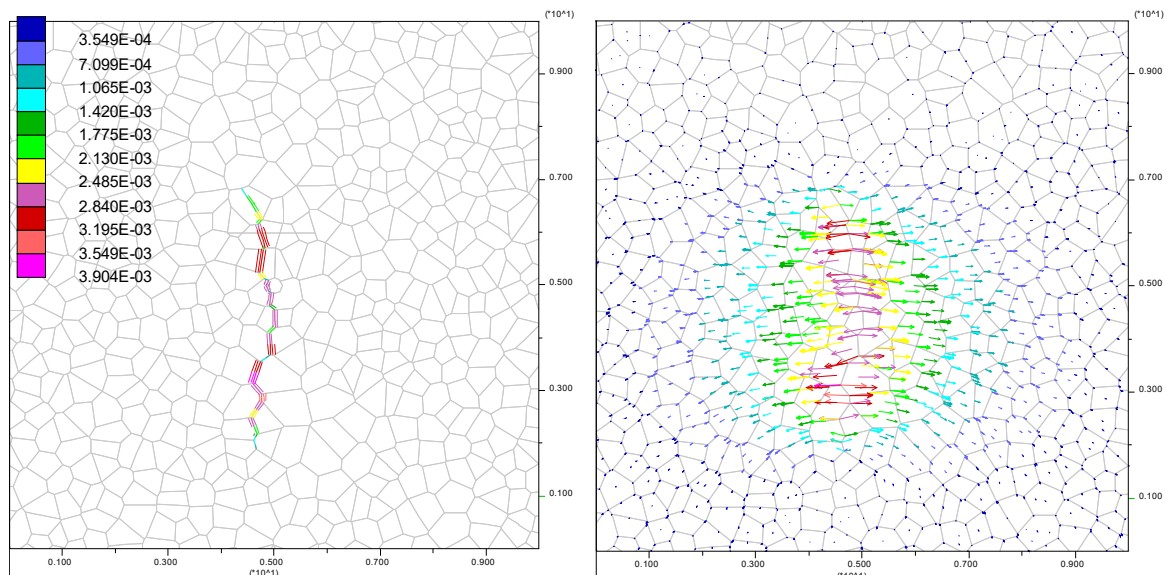

**Figure 6.** Crossing mode in numerical modeling with an incidence angle of 90° under differential stresses of 15 MPa.

### 3.3. Comparison of Different Methods

The boundary of three regions of interaction modes was determined based on numerical simulation results described in the previous sections. Based on these results, we identified a clear region of opening and crossing interactions modes, separated by a region of arresting interaction (Figure 7). Then, these results were compared with two analytical methods of Blanton [13] and Sarmadivaleh–Rasouli [16], and also the experimental results of Zhou et al. [7]. Figure 8 shows the crossed boundary of different methods. Two boundaries of maximum and minimum of "b value" in Blanton method are considered. Using Equation (2) The $bm_{in}$ value was considered 0 and $b_{max}$ value was calculated to be 0.256.

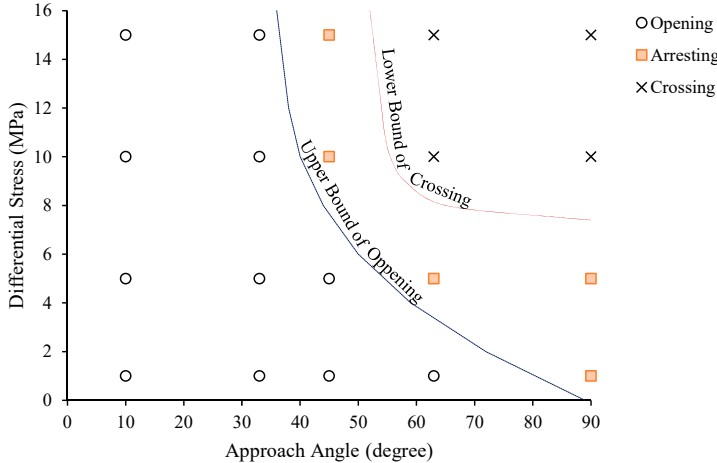

**Figure 7.** Correlation between approach angle and differential stress, based on discrete element method (DEM) simulations.

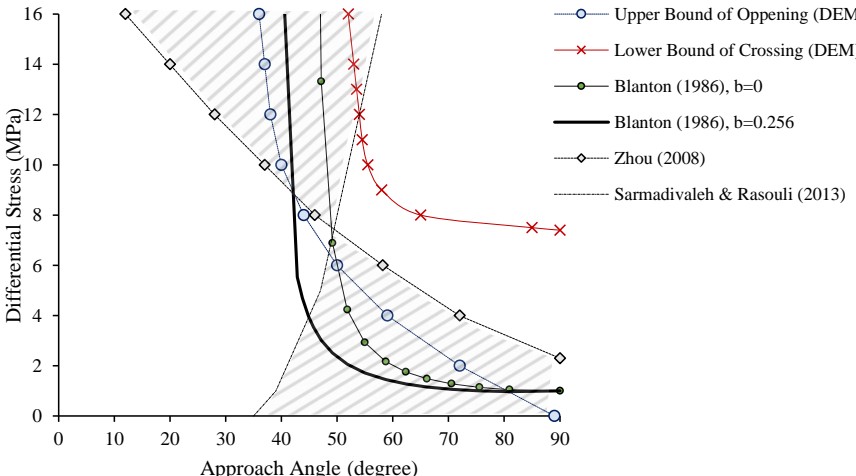

**Figure 8.** Comparison crossing bound in different methods.

The main difference of these methods is shown as a hatched area in Figure 8. As can be seen, in Sarmadivaleh and Rasouli's [16] method crossing mode occurs in a medium to high approach angle (more than 45°) even in the very low differential stress while crossing mode requires a minimum differential stress in the rest methods. Regardless of the difference in these methods, it can generally be said that HF tends to cross the NF at an angle of more than 45° and a moderate differential stress (greater than 5 MPa); and the opening mode is dominated at angles less than 45°.

## 4. Effect of Strength Parameters of Natural Fracture (NF) on HF Propagation

Cohesion (C) and friction angle (ϕ) are two parameters of fractures that seriously affect the HF. According to numerical simulation results, these parameters have two effects on HF:

- changing the interaction mode
- increasing/decreasing the joint opening

Figure 9 shows the effect of friction angle on the HF propagation in the case of an incidence angle of 33° under moderate differential stresses (Δσ = 10 MPa). As can be seen, interaction mode changes from opening to arresting mode and from arresting to crossing mode as ϕ increases from 5° to 70°. This phenomenon happens because NF and rock act as a unit against fracture propagation with increasing of the friction coefficient. Indeed, with an increasing friction coefficient, more pressure is needed to open the NF. Finally, HF tends to propagate in the same initial direction (perpendicular to minimum stress).

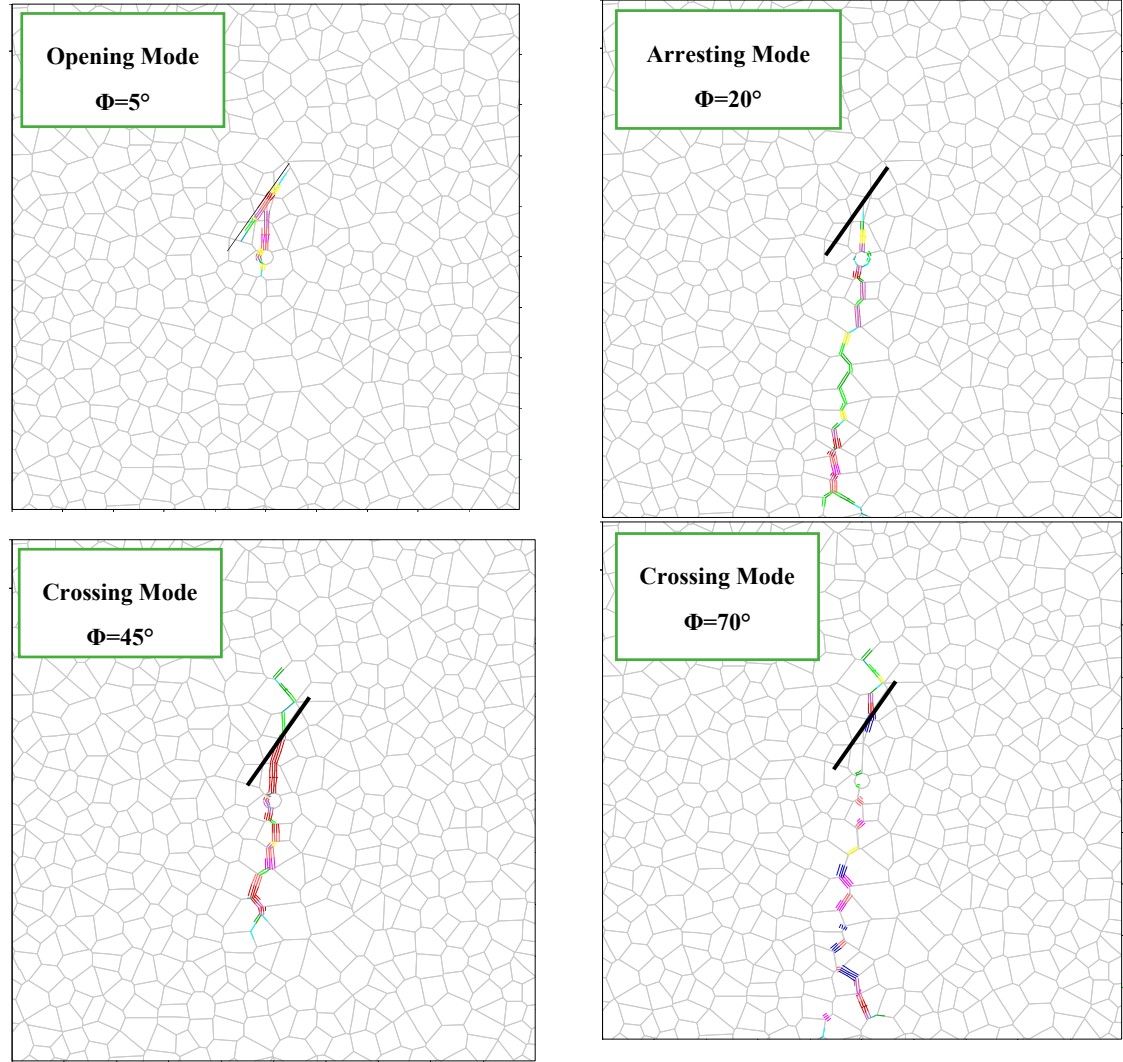

**Figure 9.** Effect of friction angle on the HF propagation (approach angle of 33° and Δσ of 10 MPa).

Changing mode was not observed with increase in C values of NF in numerical simulations. Figure 10 demonstrates the effect of cohesion on HF in the condition of an incidence angle of 33° under differential stresses of 10 MPa. This effect can be found in NF values. By increasing the C values, the

resistance of the joint to the opening is increased by the injection pressure; therefore, NF is less opened under a constant value of injection pressure (or constant flow rate over the same time period).

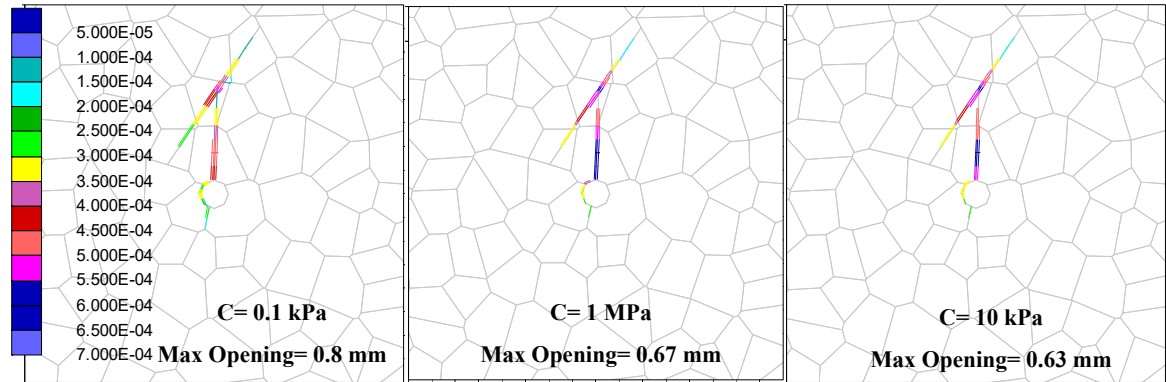

**Figure 10.** Effect of cohesion strength of NF on the HF propagation (approach angle of 33° and Δσ of 10 MPa).

Changing the interaction mode can be observed in the analytical method of Sarmadivaleh and Rasouli [16]. Figure 11 depicts the effect of strength parameters of NF on the crossed tendency. According to this figure, crossed tendency moves to the left side of the chart, leading to widening of the crossed zone. This effect is also observed in the numerical simulation results with an increase in ϕ. In this regard, the effect of ϕ is more tangible than C based on Figure 11.

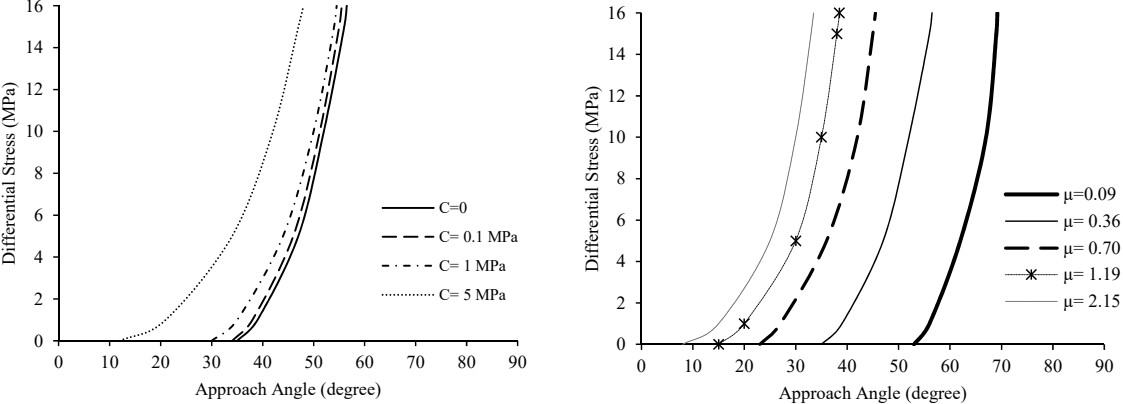

**Figure 11.** Effect of strength parameters of NF on crossed tendency based on Sarmadivaleh and Rasouli [16] method.

## 5. Conclusions

Understanding the mechanisms of interaction between NF and HF plays a key role in explaining the fracture complexity during HF treatments and, ultimately, predict fracture geometry and reservoir production. In this paper, different numerical simulations based on DEM were performed for investigating these interaction mechanisms and the effect of strength parameters of NF on HF propagation route. The numerical results also were compared with the different analytical method and experimental results. The results showed that:

- Overall, HF tends to cross the NF at an angle of more than 45° and a moderate differential stress (greater than 5 MPa); and the opening mode is dominated at an angle less than 45°.
- Interaction mode changes from opening to arresting mode and from arresting to crossing mode with an increase in ϕ from 5° to 70°.
- By increasing the amount of C, the resistance of the joint to the opening is increased by the injection pressure; therefore, NF is less opened under a constant value of injection pressure.

**Author Contributions:** Validation, R.B., K.G.; investigation, R.B.; writing—original draft preparation, R.B.; writing—review and editing, K.G. and M.A.; visualization, R.B.; supervision, K.G.; project administration, K.G. and M.A.

**Funding:** This research received no external funding.

**Conflicts of Interest:** The authors declare no conflict of interest.

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
