# Peer review of "Discrete Element Simulation of Interaction between Hydraulic Fracturing and a Single Natural Fracture"

_fluids, doi:10.3390/fluids4020076_

Round 1

Reviewer 1 Report

·         Lines 24 and 25: Preexisting NF is different than faults, joints, and bedding planes. Past studies have showed that there are clear distinctions between NF and bedding planes (or faults). Please modify the manuscript accordingly. The following references can help the authors to define the NF system in unconventional reservoirs.

o   Gale, J. F., Laubach, S. E., Olson, J. E., Eichhubl, P., & Fall, A. (2014). Natural fractures in shale: A review and new observations. AAPG bulletin, 98(11), 2165-2216.

o   Zolfaghari, A., Dehghanpour, H., Noel, M., & Bearinger, D. (2016). Laboratory and field analysis of flowback water from gas shales. Journal of unconventional oil and gas resources, 14, 113-127.

·         Figure 1: The quality of the texts on the figure should be improved. Also, please use similar font size for all text in the figure.

·         The parameters for all equations should be defined after the equations.

·         Line 59-60: … in the form “of” a single…

·         It is very hard to follow the math, mainly because none of the parameters in defined. Authors need to define each parameter when they first appear in the paper. Furthermore, having a nomenclature section at the end of the paper facilitates reading of the paper for the readers.

·         It is not common to introduce equations in the introduction section, unless it is inevitable. I recommend the authors to move equations 1-8 to another section of the paper. The main purpose of this paper is fracture network characterization. However, the introduction section lacks the literature on other characterization methods. The paper quality can be improved by introducing other fracture characterization techniques such as RTA, PTA, microseismic analysis, flowback chemical analysis, DFN model. Following references can help the authors to improve the quality of the introduction section:

o   RTA and PTA analysis for fracture characterization

§  Blanton, T. L. (1986, January). Propagation of hydraulically and dynamically induced fractures in naturally fractured reservoirs. In SPE unconventional gas technology symposium. Society of Petroleum Engineers.

§  Blanton, T. L. (1986, January). Propagation of hydraulically and dynamically induced fractures in naturally fractured reservoirs. In SPE unconventional gas technology symposium. Society of Petroleum Engineers.

o   Microseismic analysis

§  Xu, Y., Ezulike, O. D., Zolfaghari, A., Dehghanpour, H., & Virues, C. (2016, September). Complementary Surveillance Microseismic and Flowback Data Analysis: An Approach to Evaluate Complex Fracture Networks. In SPE Annual Technical Conference and Exhibition. Society of Petroleum Engineers.

§  Maxwell, S. C., Urbancic, T. I., Steinsberger, N., & Zinno, R. (2002, January). Microseismic imaging of hydraulic fracture complexity in the Barnett shale. In SPE annual technical conference and exhibition. Society of Petroleum Engineers.

o   Flowback chemical analysis

§  Zolfaghari, A., Dehghanpour, H., Ghanbari, E., & Bearinger, D. (2016). Fracture characterization using flowback salt-concentration transient. SPE Journal, 21(01), 233-244.

§  Zolfaghari, A., Dehghanpour, H., & Bearinger, D. (2019). Produced Flowback Salts vs. Induced-Fracture Interface: A Field and Laboratory Study. SPE Journal.

§  Roshan, H., Sarmadivaleh, M., & Iglauer, S. (2016). Shale fracture surface area measured by tracking exchangeable cations. Journal of Petroleum Science and Engineering, 138, 97-103.

o   DFN modeling

§  Meyer, B. R., & Bazan, L. W. (2011, January). A discrete fracture network model for hydraulically induced fractures-theory, parametric and case studies. In SPE hydraulic fracturing technology conference. Society of Petroleum Engineers.

§  Rogers, S., Elmo, D., Dunphy, R., & Bearinger, D. (2010, January). Understanding hydraulic fracture geometry and interactions in the Horn River Basin through DFN and numerical modeling. In Canadian Unconventional Resources and International Petroleum Conference. Society of Petroleum Engineers.

§  Dershowitz, W. S., Cottrell, M. G., Lim, D. H., & Doe, T. W. (2010, January). A discrete fracture network approach for evaluation of hydraulic fracture stimulation of naturally fractured reservoirs. In 44th US rock mechanics symposium and 5th US-Canada rock mechanics symposium. American Rock Mechanics Association.

·         Please specify the reasons for selection of different parameters in Table 1.

·         Hydraulic fracturing commonly happens in multi-stages (i.e., multi-stage HF). The fracturing treatment usually happens from the toe to heel of the well. How the initial fracturing treatment stages do impacts the propagation of the subsequent fracturing stages?

Author Response

Response to reviewer

Thanks a lot for reviewing our paper and presenting valuable comments to improve the quality of the paper.

The manuscript was intensively edited based on the reviewer comments. The changes are highlighted by yellow color as well as tracking change option in the Word software in the “fluids-468143-Tracking change-Re1” file. The clean version was also provided in the “fluids-468143-clean version-Re1” file. These changes are as follows:

i)            Introduction was completely revised and more relevant articles (based on reviewer suggestions) were added.

ii)                  The structure of paper was changed and some paragraphs were moved to the most appropriate place.

iii)        More details about the numerical method were added in the paper.

Comments and Suggestions for Authors

(1) Lines 24 and 25: Preexisting NF is different than faults, joints, and bedding planes. Past studies have showed that there are clear distinctions between NF and bedding planes (or faults). Please modify the manuscript accordingly. The following references can help the authors to define the NF system in unconventional reservoirs.

o   Gale, J. F., Laubach, S. E., Olson, J. E., Eichhubl, P., & Fall, A. (2014). Natural fractures in shale: A review and new observations. AAPG bulletin, 98(11), 2165-2216.

o   Zolfaghari, A., Dehghanpour, H., Noel, M., & Bearinger, D. (2016). Laboratory and field analysis of flowback water from gas shales. Journal of unconventional oil and gas resources, 14, 113-127.

The sentence and paragraph was changed.

(2)  Figure 1: The quality of the texts on the figure should be improved. Also, please use similar font size for all text in the figure.

The quality of text was improved in Figure 2.

(3) The parameters for all equations should be defined after the equations.

All parameters were defined after the equations.

(4) Line 59-60: … in the form “of” a single…

The sentence was removed and paragraph was revised.

(5) It is very hard to follow the math, mainly because none of the parameters in defined. Authors need to define each parameter when they first appear in the paper. Furthermore, having a nomenclature section at the end of the paper facilitates reading of the paper for the readers.

All parameters were defined after the equations.

(6) It is not common to introduce equations in the introduction section, unless it is inevitable. I recommend the authors to move equations 1-8 to another section of the paper. The main purpose of this paper is fracture network characterization. However, the introduction section lacks the literature on other characterization methods. The paper quality can be improved by introducing other fracture characterization techniques such as RTA, PTA, microseismic analysis, flowback chemical analysis, DFN model. Following references can help the authors to improve the quality of the introduction section:

o   RTA and PTA analysis for fracture characterization

§  Blanton, T. L. (1986, January). Propagation of hydraulically and dynamically induced fractures in naturally fractured reservoirs. In SPE unconventional gas technology symposium. Society of Petroleum Engineers.

§  Blanton, T. L. (1986, January). Propagation of hydraulically and dynamically induced fractures in naturally fractured reservoirs. In SPE unconventional gas technology symposium. Society of Petroleum Engineers.

o   Microseismic analysis

§  Xu, Y., Ezulike, O. D., Zolfaghari, A., Dehghanpour, H., & Virues, C. (2016, September). Complementary Surveillance Microseismic and Flowback Data Analysis: An Approach to Evaluate Complex Fracture Networks. In SPE Annual Technical Conference and Exhibition. Society of Petroleum Engineers.

§  Maxwell, S. C., Urbancic, T. I., Steinsberger, N., & Zinno, R. (2002, January). Microseismic imaging of hydraulic fracture complexity in the Barnett shale. In SPE annual technical conference and exhibition. Society of Petroleum Engineers.

o   Flowback chemical analysis

§  Zolfaghari, A., Dehghanpour, H., Ghanbari, E., & Bearinger, D. (2016). Fracture characterization using flowback salt-concentration transient. SPE Journal, 21(01), 233-244.

§  Zolfaghari, A., Dehghanpour, H., & Bearinger, D. (2019). Produced Flowback Salts vs. Induced-Fracture Interface: A Field and Laboratory Study. SPE Journal.

§  Roshan, H., Sarmadivaleh, M., & Iglauer, S. (2016). Shale fracture surface area measured by tracking exchangeable cations. Journal of Petroleum Science and Engineering, 138, 97-103.

o   DFN modeling

§  Meyer, B. R., & Bazan, L. W. (2011, January). A discrete fracture network model for hydraulically induced fractures-theory, parametric and case studies. In SPE hydraulic fracturing technology conference. Society of Petroleum Engineers.

§  Rogers, S., Elmo, D., Dunphy, R., & Bearinger, D. (2010, January). Understanding hydraulic fracture geometry and interactions in the Horn River Basin through DFN and numerical modeling. In Canadian Unconventional Resources and International Petroleum Conference. Society of Petroleum Engineers.

§  Dershowitz, W. S., Cottrell, M. G., Lim, D. H., & Doe, T. W. (2010, January). A discrete fracture network approach for evaluation of hydraulic fracture stimulation of naturally fractured reservoirs. In 44th US rock mechanics symposium and 5th US-Canada rock mechanics symposium. American Rock Mechanics Association.

Thanks for your comment and suggested papers. The structure of paper was modified and all suggested paper were studied. The more relative papers were added in the introduction and references.

(7) Please specify the reasons for selection of different parameters in Table 1.

In this paper two fracture systems are considered: a) fictitious joint as called Voronoi which is unreal and it has equaled properties with intact rock (because a limiting factor in the modeling stress-induced fracturing using UDEC is that all potential fracture pathways must be pre-defined), and b) natural fracture with real properties (C, φ, and …). More detail was added in the section 3.1.

(8) Hydraulic fracturing commonly happens in multi-stages (i.e., multi-stage HF). The fracturing treatment usually happens from the toe to heel of the well. How the initial fracturing treatment stages do impacts the propagation of the subsequent fracturing stages?

Thanks for reviewing the paper carefully. Hence, the aim of this paper was investigation of the effect of a single natural fracture on the hydraulic fracturing process. Therefore, the fluid flow was considered by a constant flow rate at the wellbore. Finally, different interaction modes between hydraulic fracturing and natural fracture including opening, arresting, and crossing are happened, because of the different properties of these two fracture systems as well as approaching angle and in-situ stress conditions.

Thanks in advance again for reviewing of the paper.

Sincerely,

Authors

Reviewer 2 Report

Following items should be addressed;

Novelty of this work is not clear. What does this submission add to current understanding.

Tittle should be modified to reflect that, this submission is based on numerical simulation. 

Fundamentals behind the software used for the simulation have not been explained thoroughly. 

Author Response

Response to reviewer

Thanks a lot for reviewing our paper and presenting valuable comments to improve the quality of the paper.

The manuscript was intensively edited based on the reviewer comments. The changes are highlighted by yellow color as well as tracking change option in the Word software in the “fluids-468143-Tracking change-Re1” file. The clean version was also provided in the “fluids-468143-clean version-Re1” file. These changes are as follows:

i)                    Introduction was completely revised and more relevant articles were added.

ii)                  The structure of paper was changed and some paragraphs were moved to the most appropriate place.

iii)        More details about the numerical method were added in the paper.

Comments and Suggestions for Authors

Following items should be addressed;

(1) Novelty of this work is not clear. What does this submission add to current understanding.

The purpose of this paper was investigating the effective parameters of on hydraulic fracturing. The few papers were discussed about the effect of strength parameters of natural fractures on the hydraulic fracturing. In the presented paper, this effect was clearly indicated; and a comprehensive study about the effective parameters were performed using numerical simulation and available analytical methods. 

Tittle should be modified to reflect that, this submission is based on numerical simulation.

The title was modified based on this comment.

Fundamentals behind the software used for the simulation have not been explained thoroughly. 

Thanks for your comment. The structure of the paper was changed and more detail about the numerical methodology of the paper was added in the section 3.1.

Thanks in advance again for reviewing of the paper.

Sincerely,

Authors

Round 2

Reviewer 1 Report

I am convinced with the authors answers and modifications. I recommend publication of the paper in present form.

Reviewer 2 Report

Authors have addressed my previous comments and the paper is now ready for publication.